# ENHANCING ZERO-SHOT LLM RECOMMENDATIONS VIA SEMANTICS AND COLLABORATIVE SIGNALS

## ABSTRACT

Large Language Models (LLMs) have demonstrated strong performance in rank-ing *small* candidate sets. However, when used *without any task-specific train-ing*, they still fall short compared to well-trained conventional recommender mod-els (CRMs) and fine-tuned LLM variants. We propose **SCSRec** (**S**emantic and **C**ollaborative **S**ignal-enhanced **Rec**ommendation), a *training-free* framework that bridges this performance gap through three key components: (1) an off-the-shelf LLM that transforms item features and user behaviors into rich textual repre-sentations; (2) a multi-view semantic retriever (user–user, item–item, user–item) that assembles a diverse and relevant candidate pool; and (3) a heuristic ranking prompt that incorporates CRM predictions, allowing the LLM to combine col-laborative priors with semantic reasoning. Extensive experiments on three pub-lic benchmarks and an industrial dataset show that SCSRec consistently outper-forms both established prompting baselines and fine-tuned LLM recommenders, all without any additional training overhead. Our results demonstrate that prompt engineering, when enhanced with semantics and collaborative signals, provides a competitive and cost-effective alternative to model fine-tuning for real-world recommendation tasks.

## 1 INTRODUCTION

Large Language Models (LLMs) have recently attracted significant attention in the field of recom-mender systems (RS). While LLMs excel at understanding and generating natural language, they are inherently optimized for text generation rather than recommendation tasks. As a result, directly prompting an LLM to produce recommendations is often: *(i)* computationally prohibitive, especially for large-scale datasets; *(ii)* susceptible to hallucination (Huang et al., 2025), i.e., recommending non-existent items; and *(iii)* constrained by the limited length of the prompt context.

To address these challenges, most contemporary studies (Ma et al., 2023; Fan, 2024; Wu et al., 2024; Lin et al., 2025; Liu et al., 2025) adopt a two-stage architecture: (1) a matching or pre-ranking stage, where a retriever model selects a subset of potentially relevant items, followed by (2) a ranking stage, in which LLMs reorder these candidates to better fit user intent (Li et al., 2019; Covington et al., 2016). The advanced semantic understanding and contextual reasoning capabilities of LLMs have shown considerable promise in improving the accuracy of ranking stage (Hou et al., 2024).

However, LLMs inherently lack domain-specific collaborative knowledge, as their pre-training cor-pora do not include user-item interaction data. Recent research has explored prompt engineering techniques, such as in-context learning (ICL) and chain-of-thought (CoT), to guide LLMs in rec-ommendation tasks, resulting in improvements over weak baselines (Yang et al., 2024; Yue et al., 2025; Li et al., 2024). Nevertheless, empirical evidence indicates that these methods still fall short of matching the performance of state-of-the-art conventional recommender models (CRMs) such as SASRec (Kang & McAuley, 2018) on public recommendation benchmarks (Liu et al., 2023; Wang & Lim, 2024; Hou et al., 2024; Lin et al., 2025). To address this performance gap, researchers have explored fine-tuning strategies: Luo et al. (Luo et al., 2024) introduced RecRanker, which applies instruction tuning to LLMs for diverse ranking tasks, while Bao et al. (Bao et al., 2023) proposed TALLRec, employing parameter-efficient adaptation techniques such as LoRA (Hu et al., 2022) for point-wise ranking. Although these fine-tuning approaches yield promising results (Luo et al., 2024; Bao et al., 2023; Hu et al., 2022; Yang et al., 2023; Lai et al., 2024), they introduce significant

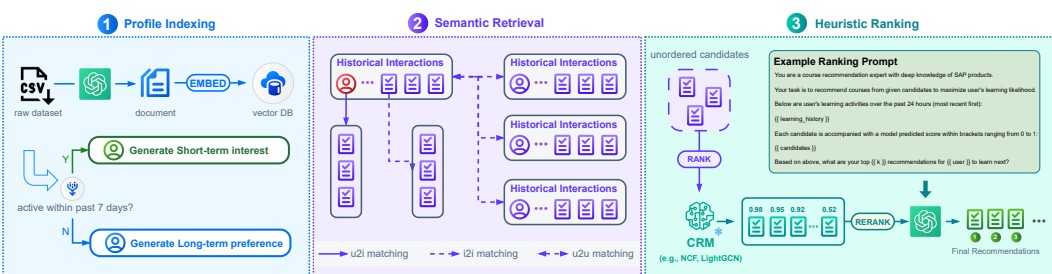

Figure 1: Overview of the proposed **SCSRec** framework.

training overhead and typically necessitate periodic retraining after deployment to accommodate evolving user preferences and newly introduced items. ously

In this paper, we propose **SCSRec** (**S**emantic and **C**ollaborative **S**ignal-enhanced **Rec**ommendation), a *training-free* framework that augments the two-stage recommendation pipeline with semantics and collaborative signals. As illustrated in Figure 1, SCSRec consists of three primary stages. For the indexing stage, we leverage LLMs to generate comprehensive item profiles based on their pre-trained knowledge and available item features. We then adaptively construct user profiles that distinguish between short-term and long-term interests by analyzing temporal interactions. These textual profiles are mapped to dense vector representations using embedding models and stored in a vector database. During the matching stage, we implement a multi-view retrieval strategy: user-user view to retrieve items consumed by similar users, item-item view to find items related to recent interactions, and user-item view to match items to the overall user profile. In the final ranking stage, we employ a CRM to produce a preliminary ranked list with associated predicted scores, which serve as collaborative signals to guide LLMs in reranking these candidates. This methodology harnesses both the commonsense knowledge and contextual understanding capabilities of LLMs, while simultaneously incorporating collaborative information derived from user behaviors—all without requiring LLM fine-tuning.

Experiments on three public benchmarks and one industrial dataset demonstrate that SCSRec achieves competitive or superior performance compared to both state-of-the-art CRMs and existing LLM-based recommenders, validating the effectiveness of prompting paradigm for LLM-based recommendation.

The main contributions of this work are as follows:

- We propose a **training-free framework** that seamlessly integrates LLM-based semantic understanding with CRM-derived collaborative signals, eliminating fine-tuning requirements while maintaining adaptability to new users, items, and even domains.

- We introduce a *multi-view semantic retrieval* strategy that enhances candidate recall by leveraging user-user, item-item, and user-item similarities. To further enhance user modeling, we employ *adaptive user indexing* to construct more expressive user profiles.

- We design a *heuristic ranking prompt* that explicitly incorporates CRM-predicted scores, enabling the LLM to reason jointly over richer textual content and collaborative priors.

- We conduct extensive experiments demonstrating that SCSRec outperforms strong baselines on both public and industrial datasets.

## 2 RELATED WORK

Recent research has explored various approaches to adapt LLMs for recommendation tasks. To tackle LLMs' context limitation and the "Lost in the Middle" problem (Liu et al., 2024), researchers employ pre-filtering modules to narrow down candidate items before leveraging LLMs for ranking. Based on whether LLMs undergo parameter updates, these approaches can be categorized into tuning and non-tuning paradigms.

## 2.1 TUNING PARADIGM

To achieve superior performance when bridging the gap between LLMs' content generation capabilities and recommendation tasks, researchers have developed various parameter-tuning approaches. GPT4Rec (Li et al., 2023) employs a fine-tuned GPT-2 model to generate queries from user interaction sequences, facilitating BM25-based item retrieval. LlamaRec (Yue et al., 2023) first utilizing LRURec (Yue et al., 2024) to retrieve top-$k$ recommendations, then applying an instruction-tuned Llama2-7B to compute ranking scores via logit extraction for listwise ranking. InstructRec (Zhang et al., 2025a) introduces a unified instruction format and leverages GPT-3.5 to generate millions of instructional examples to train Flan-T5-XL, which subsequently reranks candidates identified by the matching module during inference. RecRanker (Luo et al., 2024) develops a hybrid ranking methodology using instruction-tuned LLMs that integrates pointwise, pairwise, and listwise ranking approaches. LLaRA (Liao et al., 2024) designs an innovative curriculum prompt tuning scheme that enables Llama2-7B to comprehend item representations for listwise ranking, strategically incorporating positive items with randomly sampled negative instances in candidate pools. TransRec (Lin et al., 2024) proposes a sophisticated three-facets identifier for item representation and reconstructs instruction data for LLM tuning, generating identifiers through beam search for candidate reranking. CoLLM (Zhang et al., 2025b) presents a hybrid encoding approach that integrates collaborative information from traditional collaborative models with textual information from LLMs, utilizing LoRA for parameter-efficient fine-tuning of Vicuna-7B specifically for recommendation prediction tasks.

## 2.2 NON-TUNING PARADIGM

Other researchers have explored the zero-shot (without additional training) ranking capabilities of LLMs. Dai et al. (Dai et al., 2023) conducted pioneering research that methodically evaluated ChatGPT across pointwise, pairwise, and listwise paradigms, showing that LLMs demonstrate superior performance particularly in the latter two approaches. Liu et al. (Liu et al., 2023) employed GPT-3.5-turbo for candidate ranking through sophisticated prompt engineering while utilizing BERT embeddings to map outputs to ground-truth items, concluding that exclusive reliance on ChatGPT for sequential recommendation yields suboptimal performance. Wang et al. introduced NIR (Wang & Lim, 2023) which implements user-based or item-based filtering for candidate generation, followed by a three-step prompting strategy: capturing user preferences, selecting representative interactions, and performing listwise ranking. Chat-REC (Gao et al., 2023) employs traditional recommender systems for candidate generation, then utilizes ChatGPT for reranking based on user profiles and queries. LLMRank (Hou et al., 2024) investigated how ground-truth item positioning affects ranking performance, strategically combining content-based models (BM25, BERT) with interaction-based models (BPR, GRU4Rec, SASRec) for candidate generation. LLMSRec-Syn (Wang & Lim, 2024) constructs candidate pools by aggregating ground truth items with randomly selected items and explores similar users as demonstrations for ranking. Zhuang et al. (Zhuang et al., 2024) propose a Setwise approach that improves zero-shot ranking efficiency while reducing token costs compared to pairwise ranking. ToolRec (Zhao et al., 2024) fine-tunes attribute-oriented retrieval tools based on pre-trained sequential models, leveraging LLM-determined key attributes for matching and ranking.

While the tuning paradigm offers competitive performance against CRMs compared to non-tuning paradigm, it incurs substantial computational costs for parameter updates, which may present challenges in practical deployment. Moreover, current research has not thoroughly investigated the performance degradation that may occur when fine-tuned LLM parameters become outdated over time.

# 3 METHODOLOGY

The overall architecture of SCSRec is illustrated in Figure 1. In this section, we provide a detailed exposition of the three principal stages, demonstrating how semantic knowledge integration and traditional collaborative signals collectively enhance the performance of LLM-based recommender systems.

## 3.1 INDEX PROFILE

As depicted in stage 1 of Figure 1, the foundational step of our framework is profile indexing, where we systematically represent discrete user and item features as rich textual profiles using LLMs. These textual profiles are subsequently mapped into dense vector representations via embedding models, enabling efficient retrieval in vector databases later.

**Item Indexing** We treat item profiles as objective entities, leveraging the extensive pre-training of LLMs on large-scale, diverse corpora (e.g., movies, products, games). To fully exploit the latent knowledge of LLMs, we design domain-specific prompts tailored for each domain, allowing the model to interpret item metadata such as titles, release years, or product categories without further fine-tuning. This approach enables zero-shot item profile generation that is both flexible and generalizable across domains.

Formally, for an item $i$ with metadata $\mathcal{M}_i$, we generate a textual profile $T_i$ via a prompt function $f_{\text{prompt}}(\cdot)$:

$$T_i = f_{\text{prompt}}(\mathcal{M}_i) \tag{1}$$

The textual profile $T_i$ is then encoded into a dense vector representation $\mathbf{v}_i$ using an embedding model $g(\cdot)$:

$$\mathbf{v}_i = g(T_i) \in \mathbb{R}^d \tag{2}$$

**User Indexing** Unlike items, user profiles are inherently subjective and predominately constructed from historical user-item interactions. Previous studies (Wang & Lim, 2023; Gao et al., 2023; Hou et al., 2024; Wang & Lim, 2024; Zhao et al., 2024) often represent user preferences using the latest $H$ interactions, forming an implicit *short-term* profile. Building on this, Liu et al. (Liu et al., 2025) further aggregate such slices to approximate *long-term* preferences.

However, these approaches often neglect absolute temporal information, focusing mainly on interaction sequences. Empirical evidence suggests that recent activities (e.g., purchases within the past 24 hours) are more predictive of current user intent than older actions. To address this, we propose an *adaptive user indexing* strategy to explicitly capture short-term interests or long-term preferences.

Let $\mathcal{I}_u = [i_1, i_2, \ldots, i_N]$ denote the chronological interaction history of user $u$, where each $i_j$ is an interaction record ordered by time.

**Short-term interests:** If user $u$ has at least one interaction within the past 7 days, i.e.,

$$\mathcal{S}_u = \{i_j \mid t_j \geq t_{\text{target}} - 7 \text{ days}\} \tag{3}$$

and $|\mathcal{S}_u| > 0$, we construct the short-term profile using the most recent 5 interactions (as identified by Hou et al. (Hou et al., 2024) to be optimal) from $\mathcal{S}_u$:

$$\mathcal{H}_u^{\text{short}} = \text{Top-5}(\mathcal{S}_u) \tag{4}$$

The textual profile and its embedding are then generated as:

$$T_u = f_{\text{prompt}}(\mathcal{H}_u^{\text{short}}) \tag{5}$$

$$\mathbf{v}_u = g(T_u) \in \mathbb{R}^d \tag{6}$$

**Long-term preferences:** If $|\mathcal{S}_u| = 0$, indicating no recent activity, we then construct the long-term user profile by iteratively aggregating over the entire interaction history $\mathcal{I}_u$ in chunks of size 10. Specifically, we partition $\mathcal{I}_u$ into $M = \lceil N/10 \rceil$ consecutive chunks:

$$\mathcal{C}_u^{(m)} = [i_{10(m-1)+1}, \ldots, i_{\min(10m,N)}], \quad m = 1, \ldots, M \tag{7}$$

We initialize the user profile as `None`:

$$T_u^{(0)} = \text{None} \tag{8}$$

For each chunk $m = 1$ to $M$, we update the user profile iteratively:

$$T_u^{(m)} = f_{\text{prompt}}(T_u^{(m-1)}, \mathcal{C}_u^{(m)}) \tag{9}$$

$$\mathbf{v}_u^{(m)} = g(T_u^{(m)}) \in \mathbb{R}^d \tag{10}$$

The final long-term user profile is given by $T_u^{(M)}$ and its embedding $\mathbf{v}_u^{(M)}$.

This adaptive, iterative procedure allows LLM to flexibly compress and summarize user interaction histories, yielding robust representations even for users with sparse or irregular activities.

## 3.2 Semantic Retrieval

Existing studies (Yue et al., 2023; Gao et al., 2023; Luo et al., 2024; Hou et al., 2024) utilize CRMs for candidate generation, which effectively leverage user-item interaction signals but insufficiently address semantic relationships between users and items. Inspired by the Retrieval-Augmented Generation (RAG) (Lewis et al., 2020), we propose a multi-view retrieval which augments candidate sets with semantically retrieved items, enhancing diversity and relevance.

As illustrated in stage 2 of Figure 1, SCSRec employs three complementary semantic retrieval views:

**User–User View** We identify the $k$ most similar users to the target user $u$ by computing the cosine similarity between their embeddings:

$$\text{Sim}(u, u') = \frac{\mathbf{v}_u \cdot \mathbf{v}_{u'}}{\|\mathbf{v}_u\|\|\mathbf{v}_{u'}\|} \tag{11}$$

For each similar user $u'$, we select up to $r$ of their most recent interacted items (excluding those already interacted with by $u$) to expand the candidate pool. Here, $r$ is determined dynamically based on user temporal activity:

$$r = \begin{cases} \min\left(r_{\max}, \, n_{\text{7-day}}\right) & \text{if } n_{\text{7-day}} > 0 \\ r_{\max} & \text{otherwise} \end{cases} \tag{12}$$

where $r_{\max}$ is a predefined maximum (e.g., 5), and $n_{\text{7-day}}$ denotes the number of interactions within the past 7 days.

**Item–Item View** For the target user's recent interaction history, we consider the most recent $r$ items, denoted as $\{i_1, \ldots, i_r\}$, where $r$ is dynamically set as described above. For each item $i_j$, we retrieve the $k$ most similar items based on cosine similarity between item embeddings:

$$\text{Sim}(i, i') = \frac{\mathbf{v}_i \cdot \mathbf{v}_{i'}}{\|\mathbf{v}_i\|\|\mathbf{v}_{i'}\|} \tag{13}$$

This view enriches the candidate set with items semantically close to the user's most recent interests, promoting timely and relevant recommendations.

**User–Item View** In addition to user-user and item-item matches, we further compute similarity between the user's profile embedding $\mathbf{v}_u$ and all item embeddings $\mathbf{v}_i$, retrieving the top-$k$ most similar items:

$$\text{Sim}(u, i) = \frac{\mathbf{v}_u \cdot \mathbf{v}_i}{\|\mathbf{v}_u\|\|\mathbf{v}_i\|} \tag{14}$$

This view ensures that items highly aligned with the user's overall profile are included in the candidate set.

The multi-view semantic retrieval process merges the results via Reciprocal Rank Fusion (RRF). The overall computational complexity is $O(kr)$, since the first two views each require at most $k$ similarity computations over $r$ recent interactions, while the latter view introduces only $\mathcal{O}(k)$ additional computations.

## 3.3 Heuristic Ranking

Recent findings by Hou *et al.* (Hou et al., 2024) reveal that the positional ordering of candidates can significantly impact the zero-shot ranking capabilities of LLMs. While the common practice is to simply pass the retrieval order to the LLM for re-ranking, this procedure discards the rich collaborative information already captured by CRMs.

Motivated by Prophet (Shao et al., 2023), we further refine the ranking stage by explicitly incorporating CRM-predicted scores and the ordered candidate set to prompt LLM. This integration introduces informative auxiliary signals that substantially improve the LLM ranking accuracy.

**Candidate Construction**   Given a user $u$, let $\mathcal{C}$ denote the semantically retrieved candidates and $\mathcal{T} = \mathrm{CRM}(u, k)$ denote the top-$k$ items recalled by the CRM from the entire item corpus. The unified candidate pool is then

$$\mathcal{C}_u = \mathcal{C} \cup \mathcal{T}. \tag{15}$$

**Incorporating Collaborative Signals**   A CRM trained for click-through rate (CTR) prediction yield probability scores $p_{u,i} \in [0, 1]$ for each user-item pair $(u, i)$, indicating the predicted likelihood of user $u$ interacting with item $i$:

$$p_{u,i} = \mathrm{CRM}(u, i) \tag{16}$$

Each candidate item $i \in \mathcal{C}_u$ is assigned its corresponding predicted score $p_{u,i}$. The scored candidate set can thus be represented as:

$$\mathcal{C}_u^{\mathrm{scored}} = \left\{ (i, p_{u,i}) \mid i \in \mathcal{C}_u \right\}, \tag{17}$$

**LLM-based Ranking**   To provide richer context, we segment each user's most recent $r$ interactions into temporal bins (e.g., past 24h, 7d, 1M, earlier), denoted as $\mathcal{H}_u^{\mathrm{temporal}} = \{\mathcal{B}_u^{(m)}\}_{m=1}^M$. This allows the LLM to distinguish between recent and distant behaviors when reranking. Given the scored candidate set $\mathcal{C}_u^{\mathrm{scored}}$ and temporal history $\mathcal{H}_u^{\mathrm{temporal}}$, the LLM is prompted to generate a ranked list $\pi_u$:

$$\pi_u = \mathrm{LLM}\left(\mathcal{H}_u^{\mathrm{temporal}}, \mathcal{C}_u^{\mathrm{scored}}\right). \tag{18}$$

## 4   EXPERIMENT

In this section, we present a comprehensive set of experiments designed to address the following research questions:

- **RQ1**: Does the proposed SCSRec outperform existing baseline models in terms of recommendation performance?
- **RQ2**: How does varying the length of recent user interaction history ($r$) affect ranking performance?
- **RQ3**: How does the choice of LLM impact the final performance of the proposed approach?
- **RQ4**: What is the effect of key components on the overall recommendation effectiveness?

### 4.1   SETTINGS

**Datasets.**   We evaluate SCSRec on four datasets from different domains: Movielens-100k (Harper & Konstan, 2015), Amazon **Beauty** and **Digital Music** (He & McAuley, 2016; McAuley et al., 2015), and a real-world industrial dataset **SAP Learning (LSC)**. For all datasets, we apply 20-core filtering to retain active users. Detailed statistics are provided in Appendix B.1.

**Metrics.**   We adopt standard metrics in recommender systems: Hit Ratio (HR) and Normalized Discounted Cumulative Gain (NDCG), both at cutoff $k = 3$, which balances efficiency and comparability with prior LLM-based recommendation work. Formal definitions are deferred to Appendix B.2.

**Baseline Models.**   We compare SCSRec with three conventional recommenders (NCF (He et al., 2017), LightGCN (He et al., 2020), SASRec (Kang & McAuley, 2018)) and two representative LLM-based methods (P5 (Geng et al., 2022; Xu et al., 2024) and LLMRank (Hou et al., 2024)). Model details and training configurations are summarized in Appendix B.3.

**Implementation.**   We follow leave-one-out evaluation (Kang & McAuley, 2018; Hou et al., 2024) with all-item ranking. For LLM inference, we fix randomness (temperature $= 0$, $top\_p = 1$, seed $= 2025$) and use GPT-4o-mini with `text-embedding-3-large`. Other hyperparameters (e.g., history length $r = 5$) and training setups for P5 and LLMRank are described in Appendix B.4.

## 4.2 Main Results (RQ1)

Table 1 compares SCSRec with baselines across four datasets. Several key observations emerge.

First, SCSRec achieves consistent improvements over both base CRMs (NCF, LightGCN, SAS-Rec) and prompting-based LLMRank. For example, SCSRec$_{\text{LightGCN}}$ attains the best HR@3 and NDCG@3 on ML-100K, while SCSRec$_{\text{SASRec}}$ leads on Beauty, Music, and LSC. These gains highlight the method's effectiveness and generalizability across domains and model architectures.

Second, base CRMs generally outperform LLMRank. On ML-100K, for instance, NCF, LightGCN, and SASRec all exceed their LLMRank counterparts, reflecting prior findings that prompting-based re-ranking lags behind well-trained CRMs.

Notably, the performance of prompting based zero-shot recommendation approaches is highly contingent on the strength of the underlying CRM. When paired with a powerful base model, SCSRec consistently outperforms the fine-tuning based OpenP5 across most datasets. Moreover, in certain cases, even LLMRank surpasses OpenP5 (e.g., on ML-100K and Music), indicating that prompting-based LLM-RS can be competitive with, and even superior to, fine-tuning-based solutions when supported by a robust CRM. These results suggest that the prompting based recommendation paradigm is also worthy of in-depth investigation, despite approaches such as LLM fine-tuning.

Table 1: Performance comparison of different methods on four datasets. **Bold** indicates the best performance and underline indicates the second-best.

| Method | ML-100K | | Beauty | | Music | | LSC | |
|---|---|---|---|---|---|---|---|---|
| | HR@3 | NDCG@3 | HR@3 | NDCG@3 | HR@3 | NDCG@3 | HR@3 | NDCG@3 |
| OpenP5 | 0.0467 | 0.0343 | 0.0400 | 0.0307 | 0.0208 | 0.0150 | 0.1153 | 0.0901 |
| NCF | 0.0435 | 0.0336 | 0.0178 | 0.0147 | 0.0208 | 0.0165 | 0.0438 | 0.0325 |
| LLMRank$_{\text{NCF}}$ | 0.0392 | 0.0291 | 0.0133 | 0.0108 | 0.0243 | 0.0157 | 0.0570 | 0.0452 |
| SCSRec$_{\text{NCF}}$ | 0.0562 | 0.0425 | 0.0555 | 0.0391 | 0.0243 | 0.0191 | 0.0879 | 0.0666 |
| LightGCN | 0.0838 | 0.0661 | 0.0200 | 0.0121 | 0.0312 | 0.0205 | 0.1098 | 0.0824 |
| LLMRank$_{\text{LightGCN}}$ | 0.0530 | 0.0383 | 0.0100 | 0.0064 | 0.0278 | 0.0202 | 0.0922 | 0.0675 |
| SCSRec$_{\text{LightGCN}}$ | **0.1007** | **0.0789** | 0.0555 | 0.0391 | 0.0365 | 0.0250 | 0.1338 | 0.1031 |
| SASRec | 0.0785 | 0.0599 | 0.0466 | 0.0368 | 0.0451 | 0.0354 | 0.1603 | 0.1271 |
| LLMRank$_{\text{SASRec}}$ | 0.0583 | 0.0407 | 0.0411 | 0.0320 | 0.0330 | 0.0274 | 0.0903 | 0.0668 |
| SCSRec$_{\text{SASRec}}$ | 0.0848 | 0.0651 | **0.0777** | **0.0613** | **0.0521** | **0.0409** | **0.1730** | **0.1369** |

## 4.3 Impact of Recent Interaction History Length (RQ2)

As mentioned in previous experiment settings, we set the recent interaction history length $r$ to 5 which follows established findings from prior research (Hou et al., 2024). However, the choice of history length may significantly influence the performance of SCSRec, as it not only impacts the semantically retrieved candidate set but also shapes how the LLM interprets user behavior. To explore this aspect, we conduct a series of experiments varying the recent interaction history length to assess its impact on the effectiveness of SCSRec. Specifically, we evaluate SCSRec with different history lengths (e.g., 5, 10, 15, 20) on benchmark datasets, the experimental results can be found in Figure 2.

For ML-100K and Beauty, performance consistently declines as $r$ increases beyond 5. For instance, on ML-100K with NCF, HR@3 decreases from 0.0562 at $r = 5$ to 0.0477 at $r = 20$, while on Beauty the drop exceeds 37%. These results suggest that longer histories may introduce noise and dilute recent preferences in sparse domains.

In contrast, Music and LSC show stable performance across different $r$ values. For example, NCF on LSC remains nearly unchanged (HR@3: 0.0879 at $r = 5$ vs. 0.0876 at $r = 20$), with similar stability observed for other models. This indicates that in denser or semantically richer domains, SCSRec is less sensitive to history length.

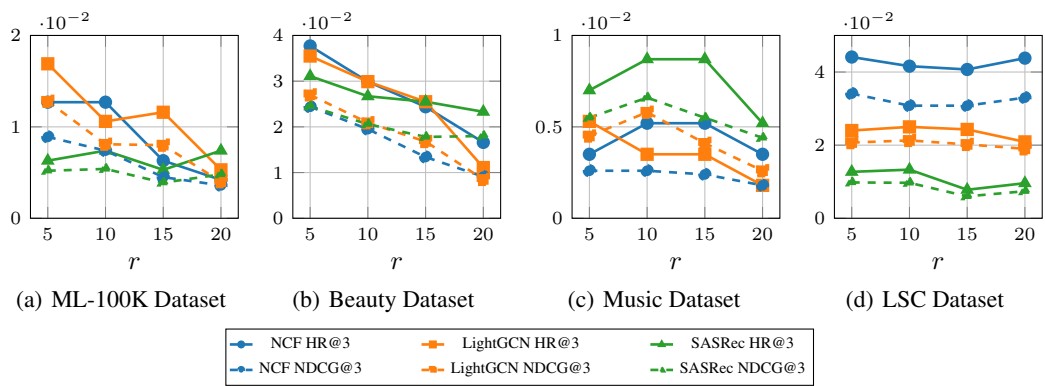

Figure 2: Absolute improvements for different $r$ over base CRMs on four datasets.

## 4.4 EFFECT OF LARGE LANGUAGE MODEL SELECTION (RQ3)

To examine whether the choice of backbone LLM affects SCSRec, we evaluated six representative models during the ranking stage: two commercial LLMs (GPT-4o-mini and Qwen-turbo) and four open-source models (Llama-3.1-8B and Qwen2.5 series: 3B, 7B, 14B). Figure 3 reports HR@3 improvements across four datasets (ML-100K, Beauty, Music, and LSC) with three baseline CRMs (NCF, LightGCN, SASRec). Several key observations emerge from our analysis:

**Commercial LLMs perform best overall.** GPT-4o-mini achieves consistently strong results, while Qwen-turbo is highly competitive and occasionally surpasses GPT-4o-mini, indicating domain-specific advantages.

**Open-source LLMs improve performance but remain less stable.** Models such as Llama-3.1-8B and Qwen2.5-7B generally enhance base CRMs, though their gains are smaller and more variable. In some cases, they match or exceed commercial models, but performance can also degrade relative to baselines.

**Scaling effects are evident.** Among Qwen2.5 series, those with more parameters generally achieve superior performance. Notably, Qwen2.5-3B sometimes performs worse than the base models, highlighting that scale is a critical factor for effectiveness in recommendation tasks.

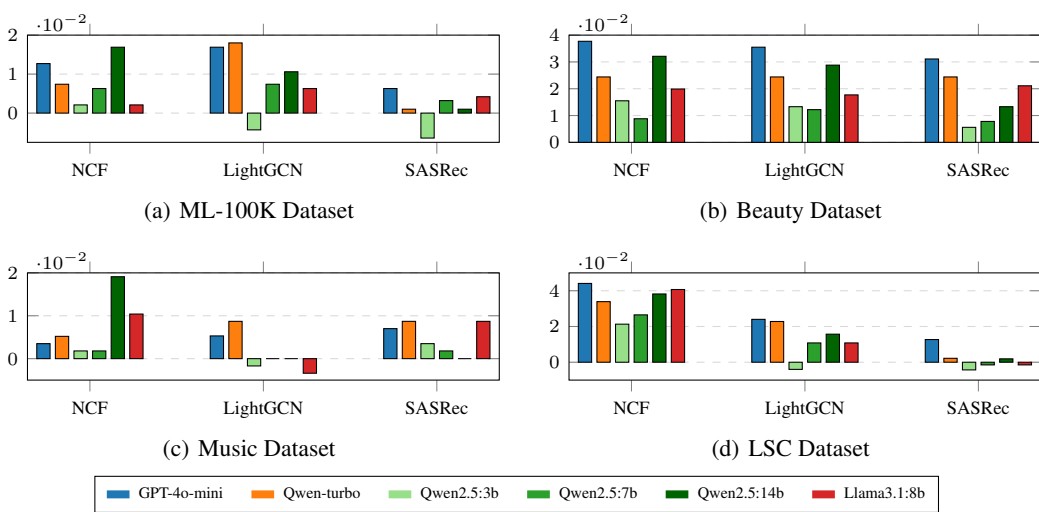

Figure 3: HR@3 absolute improvements for different LLMs over base models among four datasets.

## 4.5 ABLATION STUDY (RQ4)

We conducted ablation experiments on SCSRec using the best-performing base CRM for each dataset (LightGCN for ML-100K; SASRec for Beauty, Music, and LSC). Three variants were considered: (1) **w/o SR**, which removes the semantic retrieval module and relies solely on candidate items generated by the CRM; (2) **w/o CS**, which omits the collaborative signals, directly feeding candidates into the LLM without scoring; and (3) **w/o SR+CS**, which eliminates both SR and CS modules, aligning with the setup of most prior prompting-based studies. Results are summarized in Table 2.

**Semantic Retrieval (SR)**   SR notably improves performance, particularly on datasets with rich semantics or sparse interactions. Its removal causes a large drop on Beauty, moderate drops on ML-100K and LSC, and only minor decreases on Music, indicating its impact depends on dataset characteristics.

**Collaborative Signal (CS)**   The collaborative signal, derived from CRM scores, is vital for guiding LLM reranking, with its removal causing notable drops on LSC, ML-100K, and Music. In Beauty, however, omitting CS slightly improves results, suggesting semantics dominate there. Overall, CS is generally essential except in strongly semantic domains.

**Combination of SR and CS**   The variant without both SR and CS modules (**w/o SR+CS**) shows the lowest or second-lowest performance across all datasets. This highlights the complementary nature of semantic retrieval and collaborative signals: integrating both is essential for maximizing recommendation accuracy in diverse scenarios.

Table 2: Ablation Study Results across four datasets with their best-performing base CRM.

| Method | ML-100K | | Beauty | | Music | | LSC | |
|---|---|---|---|---|---|---|---|---|
| | HR@3 | NDCG@3 | HR@3 | NDCG@3 | HR@3 | NDCG@3 | HR@3 | NDCG@3 |
| SCSRec | **0.1007** | **0.0789** | 0.0777 | 0.0613 | **0.0521** | **0.0409** | **0.1730** | **0.1369** |
| w/o SR | 0.0930 | 0.0752 | 0.0451 | 0.0373 | 0.0515 | 0.0391 | 0.1638 | 0.1317 |
| w/o CS | 0.0870 | 0.0719 | **0.0829** | **0.0694** | 0.0498 | 0.0391 | 0.1350 | 0.1100 |
| w/o SR+CS | 0.0855 | 0.0720 | 0.0440 | 0.0363 | 0.0440 | 0.0352 | 0.1517 | 0.1232 |

## 5 CONCLUSION

In this paper, we proposed **SCSRec**, a prompt-based framework that unifies the semantic reasoning abilities of large language models (LLMs) with the collaborative knowledge of conventional recommender models (CRMs). By incorporating semantic representations and integrating collaborative signals into the prompting process, SCSRec enhances candidate diversity and overall recommendation quality. Experiments show that SCSRec delivers competitive performance without parameter tuning of the LLM, highlighting the practicality of prompt-based approaches for modern recommender systems and motivating further research in this direction.

## DISCLOSURE OF LLM USAGE

We used ChatGPT as an assistive tool during this research. The LLM was employed for the following purposes: (1) refining the clarity and conciseness of paper writing (e.g., grammar, style, and wording suggestions); (2) assisting in LaTeX formatting and visualization adjustments.

The LLM was not used for research ideation, experimental design, or generating results. All LLM-generated content was carefully reviewed and verified by the authors.

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

## A   SUPPLEMENTARY MATERIALS

### A.1   CODE REPOSITORY

To facilitate reproducibility and further research, we provide the source code associated with this paper at the following repository:

```
https://anonymous.4open.science/r/SCSRec-5176/README.md
```

The repository contains:

- Implementation of our proposed **SCSRec** framework.
- Scripts for preprocessing datasets and running experiments.
- Instructions for reproducing the results reported in the paper.

## B EXPERIMENTAL DETAILS

### B.1 DATASETS

We evaluate SCSRec on four datasets:

- **Movielens-100k** (Harper & Konstan, 2015): contains users who each provided at least 20 ratings.
- **Amazon Beauty** and **Amazon Digital Music** (He & McAuley, 2016; McAuley et al., 2015): we apply 20-core filtering to retain users with at least 20 interactions, ensuring consistency with Movielens-100k.
- **SAP Learning (LSC)**: a real-world industrial dataset of user learning activities between 2023/01/01 and 2025/01/01. We apply the same 20-core filtering, retaining only users who have initiated at least 20 distinct courses.

The detailed statistics of all datasets are listed in Table 3.

Table 3: Dataset Statistics

| Dataset | User | Item | Interactions | Sparsity |
|---------|------|------|--------------|----------|
| ML-100K | 943 | 1,682 | 100,000 | 93.70% |
| Beauty | 901 | 7,555 | 29,731 | 99.56% |
| Music | 576 | 4,507 | 25,120 | 99.03% |
| LSC | 3,243 | 1,073 | 98,697 | 97.16% |

### B.2 EVALUATION METRICS

We adopt two widely used metrics:

- **Hit Ratio (HR@k)**: measures whether the ground-truth item appears in the top-$k$ recommendations.
- **NDCG@k**: assigns higher rewards when the ground-truth item appears closer to the top of the ranking.

Formally, for a test instance with ground-truth item $i^*$, HR@k is defined as:

$$HR@k = \mathbb{I}[i^* \in \text{Top-}k],$$

and NDCG@k is defined as:

$$NDCG@k = \frac{1}{\log_2(p+1)} \quad \text{if } i^* \text{ is ranked at position } p \leq k, \quad 0 \text{ otherwise.}$$

We set $k = 3$ throughout all experiments.

### B.3 BASELINE MODELS

We compare SCSRec against both conventional and LLM-based baselines:

- **NCF** (He et al., 2017): Neural Collaborative Filtering, which leverages a multi-layer perceptron to model user-item interactions.
- **LightGCN** (He et al., 2020): A simplified graph convolutional network that learns user and item embeddings via linear propagation on the interaction graph.
- **SASRec** (Kang & McAuley, 2018): A sequential recommender based on a unidirectional Transformer to capture user behavior patterns.
- **P5** (Geng et al., 2022; Xu et al., 2024): A fine-tuning paradigm for LLM-based recommendation, where a pretrained language model is adapted to the recommendation domain through supervised training.
- **LLMRank** (Hou et al., 2024): A prompt-based recommendation method leveraging zero-shot ranking capabilities of LLMs to re-rank candidates retrieved by CRMs.

### B.4 IMPLEMENTATION DETAILS

We follow established protocols (Kang & McAuley, 2018; Hou et al., 2022; 2024; Liu et al., 2025):

- **Evaluation protocol**: leave-one-out strategy, where the most recent interaction is used for testing and the penultimate for validation. Recommendations are generated from the full set of items unseen by the user (all-item evaluation).

- **LLM settings**: GPT-4o-mini (2024-07-18) as the primary LLM, with `text-embedding-3-large` for embeddings. Randomness eliminated by setting temperature $= 0$, $top\_p = 1.0$, and random seed $= 2025$.

- **Hyperparameters**: user history length $r = 5$ for all datasets, following prior studies (Hou et al., 2024).

- **P5 training**: implemented via OpenP5 (Xu et al., 2024) with T5 backbone, sequential task formulation, sequential item indexing, and seen template setting (seen:0). Other parameters follow OpenP5 defaults.

- **LLMRank**: tested with three CRMs (NCF, LightGCN, SASRec) as retrieval models. In the ranking stage, GPT-4o-mini is used with a recency-focused prompting strategy as described in the original paper.

