# OpenReview forum: "Enhancing Zero-Shot LLM Recommendations via Semantics and Collaborative Signals"
_ICLR.cc/2026/Conference — Submitted to ICLR 2026_

### Official Review · Reviewer_SYCo · 2025-10-31

**Soundness:** 3
**Presentation:** 3
**Contribution:** 2
**Rating:** 4
**Confidence:** 4

**Summary:**

Summary:

This paper introduces SCSRec, a training-free framework for zero-shot large-language-model-based recommendation. It integrates (1) semantic representations from an off-the-shelf LLM, (2) multi-view semantic retrieval (user–user, item–item, user–item), and (3) heuristic ranking prompts incorporating collaborative signals (scores from a base CRM such as NCF, LightGCN, or SASRec). Experiments on multiple datasets show consistent improvements over both conventional recommenders and prior LLM-based approaches.

**Strengths:**

Strengths
+ The motivation is clear. The paper aims to avoid training workload of fine-tuning LLMs while eliminating un-desirable performance of prompt-only approaches.
+ The designed mythology is intuitive. The design of three components of SCSRec are well-motivated.

**Weaknesses:**

Weaknesses
- The used conventional CRM like SASRec still needs training. It’s possible that the training of SASRec is even higher than efficient fine-tuning approaches like LoRA of LLMs. It can dampen the motivation and challenge the grounding of this paper.
- Not enough LLMs-based fine-tuning methods as baselines. The only one is OpenP5. Please consider comparing more LLM-based finr-tuning approaches, like TallRec [1]
- Not any LLM-based zero-shot without relying on conventional CRM is compared, like TaxRec [2].

[1] Bao et al. TALLRec: An Effective and Efficient Tuning Framework to Align Large Language Model with Recommendation.

[2] Liang et al. Taxonomy-Guided Zero-Shot Recommendations with LLMs.

**Questions:**

Please refer to the weaknesses.

---

### Official Review · Reviewer_AyPb · 2025-10-31

**Soundness:** 3
**Presentation:** 3
**Contribution:** 2
**Rating:** 4
**Confidence:** 4

**Summary:**

This paper investigates performance enhancement in zero-shot recommendation systems based on large language models (LLMs), proposing a training-free framework named SCSRec. Its core innovation lies in integrating the semantic understanding capabilities of LLMs with collaborative signals from traditional recommendation models (CRMs) without fine-tuning the LLM. Performance gains are achieved through three key modules: First,leveraging LLMs to generate comprehensive item profiles based on their pre-trained knowledge and available item features; Second, it designs multi-view semantic retrievers (user-user, item-item, user-item) to construct diverse and relevant candidate pools; Third, it introduces heuristic ranking prompts to integrate CRM predictions, combining collaborative prior knowledge with semantic reasoning. Finally, the synergistic relationship between semantic reasoning capabilities and collaborative filtering prior has been thoroughly validated, demonstrating the proposed method's effectiveness across three public benchmark datasets and one industrial dataset.

**Strengths:**

1. This paper explores a recommendation paradigm independent of LLM fine-tuning, proposing the zero-training framework SCSRec. Through the innovative combination of “semantic retrieval + collaborative prompting,” it effectively integrates the semantic understanding capabilities of LLMs with the collaborative filtering prior of traditional recommendation models (CRMs). This approach significantly enhances the performance of zero-shot LLM recommendations without requiring large model fine-tuning, while avoiding the high costs associated with fine-tuning;
2. The adaptive user indexing strategy proposed in this paper dynamically distinguishes and models users' short-term interests and long-term preferences by combining short-term time windows (e.g., recent 7-day activity detection) with long-term iterative aggregation (block-based historical summarization). This approach captures users' immediate behavioral signals while effectively mitigating noise and interest drift issues in long-sequence histories, thereby constructing more timely and expressive user representations.

**Weaknesses:**

1. The citation highlights the pain point of LLM recommendations suggesting non-existent items, but the Methods and Experiments sections fail to explain how SCSRec mitigates this issue.
2. The baseline comparison is weak, and the experimental results lack sufficient contrast, making the conclusions unconvincing.
3. The flowchart in Figure 1 for the second phase is difficult to understand intuitively.
4. This paper emphasizes that SCSRec achieves low cost, but without experimental quantification, presenting cost savings in a table would be more persuasive.

**Questions:**

1. In multi-view retrieval, the results from the three views are fused using RRF. Have other fusion strategies (e.g., weighted averaging, learning-based fusion) been explored? Is RRF the optimal choice?
2. Could more comparisons with existing methods be included?
3. Does it also yield significant improvements on weak CRM?

---

### Official Review · Reviewer_Vs9E · 2025-10-31

**Soundness:** 1
**Presentation:** 2
**Contribution:** 1
**Rating:** 2
**Confidence:** 5

**Summary:**

This paper aims to bridge the performance gap between non–fine-tuned large language models (LLMs) and well-trained conventional recommender models on rerank tasks.
It proposes SCSRec (Semantic and Collaborative Signal-enhanced Recommendation), a training-free framework that augments user profiles by modeling both short-term and long-term user interests. The system retrieves candidate items using LLM-based embeddings and applies a conventional recommender model for preliminary ranking. Finally, it introduces a prompt-based reranking stage that leverages collaborative signals from the conventional model to guide the LLM.
The authors claim that the proposed prompt-engineering strategy offers a competitive and cost-effective alternative to model fine-tuning for real-world recommendation scenarios.

**Strengths:**

**Scope:** The paper addresses a relevant issue — how to improve recommendation performance of LLMs without fine-tuning. The focus on training-free integration is practically meaningful given current computational constraints in industrial settings.

**Weaknesses:**

**Logic Flaw and Inconsistent Motivation**: Author claim to bridge the gap of training-free LLM frameworks and on ranking small candidate sets of items. However the framework consists of multiple stages, including item embedding generation, Semantic embedding-retrieval, and a conventional recommender model are needed to inject CF signal. The framework is not training-free, as conventional recommender model. This discrepancy appears throughout the paper, particularly in the introduction, related work, and methodology sections.

**Lack of Literature reviews**: The paper proposes framework is a simple combination of existing techniques, such as LLM-based embedding Retrieval, User profile augmentation, User intention analyse, and prompt-based reranking [1,2,3,4].

**Experiments**: The paper lacks a comprehensive experimental evaluation. It does not provide comparisons with strong baselines, such as some state-of-the-art LLM-based recommenders [7,8] or conventional recommender models [5,6].


**Typos**: Lines 68 e.g., ously. Table 1, forget to Bold performance.


[1] Ren, Xubin, et al. "Representation learning with large language models for recommendation." Proceedings of the ACM web conference 2024. 2024.
[2] Lyu, Hanjia, et al. "Llm-rec: Personalized recommendation via prompting large language models." arXiv preprint arXiv:2307.15780 (2023
[3] Zhang, An, et al. "On generative agents in recommendation." Proceedings of the 47th international ACM SIGIR conference on research and development in Information Retrieval. 2024.
[4] Wang, Yu, et al. "Drdt: Dynamic reflection with divergent thinking for llm-based sequential recommendation." arXiv preprint arXiv:2312.11336 (2023).
[5] Rajput, Shashank, et al. "Recommender systems with generative retrieval." Advances in Neural Information Processing Systems 36 (2023): 10299-10315.
[6] Tanner, Carmen, et al. "Actions speak louder than words." Zeitschrift für Psychologie/Journal of Psychology (2015).
[7] Bao, Keqin, et al. "Decoding matters: Addressing amplification bias and homogeneity issue for llm-based recommendation." arXiv preprint arXiv:2406.14900 (2024).
[8] Li, Xinhang, et al. "E4srec: An elegant effective efficient extensible solution of large language models for sequential recommendation." arXiv preprint arXiv:2312.02443 (2023).

**Questions:**

Refer to the weaknesses mentioned above.

---

### Official Review · Reviewer_JNUN · 2025-11-02

**Soundness:** 1
**Presentation:** 3
**Contribution:** 1
**Rating:** 2
**Confidence:** 4

**Summary:**

This paper studies how to leverage LLMs as zero-shot rerankers to address the retraining cost and potential forgetting problems.  The authors propose SCSRec, a framework that integrates semantic retrieval and collaborative signals, allowing an LLM to perform ranking without additional training. The paper is clearly written, and the motivation—exploring training-free LLM reranking as a practical solution in recall–rerank systems—has reasonable value.

**Strengths:**

S1. The paper is clearly written and well structured, making the proposed framework and experimental setup easy to follow.
S2. Using an LLM as a training-free reranker is a valuable idea that helps reduce retraining costs in typical recall–rerank recommendation pipelines.

**Weaknesses:**

W1. Lack of meaningful baselines to verify the effectiveness of the proposed method.
Only one conventional recommender model (CRM) and one LLM-based reranker are used. The comparison with the CRM baseline is limited in meaning, since the proposed method inherently adds additional content information that CRMs do not exploit. Meanwhile, the LLMRank baseline, as a reranker, often performs worse than the CRM itself. As a result, the experiments lack strong and competitive baselines to convincingly demonstrate the true advantage of the proposed approach.

W2. Unclear motivation of the Semantic Retrieval module.
The Semantic Retrieval module’s motivation is not well justified. The design essentially adds a textual-based recall path on top of the conventional collaborative filtering recall, then feeds all candidates to the LLM for re-ranking. As the main goal is to enhance the LLM’s reranking capability, simply adding another recall path seems only marginally related to that objective.

W3.  Relatively limited technical novelty.
The method follows a familiar paradigm where LLM re-ranks the candidate items based on user history and the scores predicted by the CRM. While practical, this design offers limited technical innovation to existing LLM-based recommenders.

**Questions:**

Please address the concerns raised in the Weaknesses part.

---

### Meta-Review · Area_Chair_Vuzw · 2026-01-06

**Summary:**

Weak Comparative Significance: The experiments currently rely on a traditional recommendation model (CRM) and an LLM-based re-ranker. Comparing against the CRM is of limited significance because the proposed method utilizes additional content information that the CRM does not. Furthermore, since the LLMRank baseline (as a re-ranker) typically underperforms the CRM itself, the experiments lack strong, competitive baselines to convincingly demonstrate the proposed method's advantages.

Missing SOTA Comparisons: The paper lacks a comprehensive experimental evaluation. It fails to compare the proposed method with state-of-the-art LLM-based recommenders [7,8] or strong traditional recommendation models.

Insufficient Fine-tuning Baselines: The data regarding LLM-based fine-tuning methods is insufficient, as only OpenP5 is currently included. It is recommended to compare against more LLM-based fine-tuning methods, such as TallRec.

**Reviewer Concerns:**

The issue regarding the inclusion of more meaningful and competitive baseline models (including SOTA LLM-based recommenders, strong traditional models, and fine-tuning methods such as TallRec) remains unresolved.

**Reviewer Scores:**

2，2，4，6

---

### Decision · Program_Chairs · 2026-01-26

Reject